# Training Compute-Optimal Large Language Models

**Jordan Hoffmann**[*]     **Sebastian Borgeaud**[*]     **Arthur Mensch**[*]     **Elena Buchatskaya**

**Trevor Cai**     **Eliza Rutherford**     **Diego de Las Casas**     **Lisa Anne Hendricks**

**Johannes Welbl**     **Aidan Clark**     **Tom Hennigan**     **Eric Noland**     **Katie Millican**

**George van den Driessche**     **Bogdan Damoc**     **Aurelia Guy**     **Simon Osindero**

**Karen Simonyan**     **Erich Elsen**     **Oriol Vinyals**     **Jack W. Rae**     **Laurent Sifre**[*]

[*] Equal contributions

**DeepMind**
`(sborgeaud|amensch|sifre)@deepmind.com`

## Abstract

We investigate the optimal model size and number of tokens for training a Transformer language model under a given compute budget. We find that current large language models are significantly undertrained, a consequence of the recent focus on scaling language models whilst keeping the amount of training data constant. By training over 400 language models ranging from 70 million to over 16 billion parameters on 5 to 500 billion tokens, we find that for compute-optimal training, the model size and the number of training tokens should be scaled equally: for every doubling of model size the number of training tokens should also be doubled. We test this hypothesis by training a predicted compute-optimal model, *Chinchilla*, that uses the same compute budget as *Gopher* but with 70B parameters and 4× more more data. *Chinchilla* uniformly and significantly outperforms *Gopher* (280B), GPT-3 (175B), Jurassic-1 (178B), and Megatron-Turing NLG (530B) on a large range of downstream evaluation tasks. This also means that *Chinchilla* uses substantially less compute for fine-tuning and inference, greatly facilitating downstream usage. As a highlight, *Chinchilla* reaches a state-of-the-art average accuracy of 67.5% on the MMLU benchmark, greater than a 7% improvement over *Gopher*.

## 1 Introduction

A series of *Large Language Models* (LLMs) have recently been introduced [6, 30, 38, 48, 52], with the largest dense language models now having over 500 billion parameters. These large autoregressive transformers [53] have demonstrated impressive performance on many tasks using a variety of evaluation protocols: zero-shot generalization, few-shot training, and as a basis for fine-tuning. The compute and energy cost for training large language models is substantial [38, 52] and rises with increasing model size. In practice, the allocated training compute budget is often known in advance: practitioners have access to a certain number of accelerators for a given period of time. Since it

36th Conference on Neural Information Processing Systems (NeurIPS 2022).

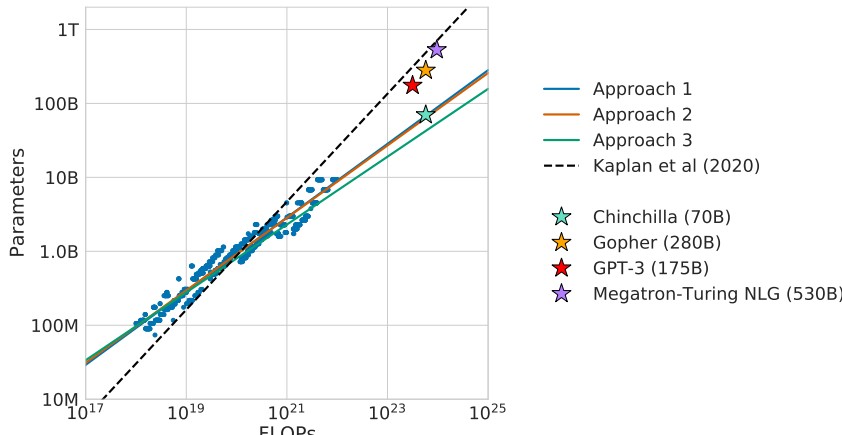

Figure 1: **Overlaid predictions.** We overlay the predictions from our three different approaches, along with projections from [23]. We find that all three methods predict that current large models should be substantially smaller and therefore trained much longer than is currently done. In Figure A3, we show the results with the predicted optimal tokens plotted against the optimal number of parameters for fixed FLOP budgets. ***Chinchilla*** **outperforms** *Gopher* **and the other large models (see Section 4.2).**

Table 1: **Current LLMs**. We show five of the current largest dense transformer models, their size, and the number of training tokens. Other than LaMDA [52], most models are trained for approximately 300 billion tokens. We introduce *Chinchilla*, a substantially smaller model, trained for much longer than 300B tokens. Table A3 shows our projected optimal relation between model size and tokens.

| Model | Size (# Parameters) | Training Tokens |
|---|---|---|
| LaMDA [52] | 137 Billion | 768 Billion |
| GPT-3 [6] | 175 Billion | 300 Billion |
| Jurassic [30] | 178 Billion | 300 Billion |
| *Gopher* [38] | 280 Billion | 300 Billion |
| MT-NLG 530B [48] | 530 Billion | 270 Billion |
| *Chinchilla* | 70 Billion | 1.4 Trillion |

is typically only feasible to train these large models once, accurately estimating the best model hyperparameters for a given compute budget is critical [51].

Kaplan et al. [23] showed that there is a power law relationship between the number of parameters in an autoregressive language model (LM) and its performance (measured in evaluation perplexity). One notable conclusion in [23] is that large models should not be trained to their lowest possible loss to be compute optimal; they argue that model size should grow faster than the size of the training set for a given increase of computational budget. As a result, the field has been training larger and larger models while keeping the size of the training set to approximately 300 billion tokens, expecting performance improvements (Table 1). While we find that there is effectively a trade-off between model size and training set size, we estimate that large models should be trained for many more training tokens than recommended by [23]. Specifically, given a $10\times$ increase computational budget we find that model size and the number of training tokens should be scaled in equal proportions.

In this work, we revisit the question: *Given a fixed FLOPs budget,[1] how should one trade-off model size and the number of training tokens?* To answer this question, we model the final pre-training loss[2] $L(N, D)$ as a function of the number of model parameters $N$, and the number of training tokens, $D$. Since the computational budget $C$ is a deterministic function FLOPs$(N, D)$ of the number of seen training tokens and model parameters, we are interested in minimizing $L$ under the constraint

---

[1]For example, knowing the number of accelerators and a target training duration.

[2]For simplicity, we perform our analysis on the smoothed training loss which is an unbiased estimate of the test loss, as the number of training tokens is less than the number of tokens in the entire corpus.

FLOPs$(N, D) = C$:

$$N_{opt}(C), D_{opt}(C) = \underset{N,D \text{ s.t. FLOPs}(N,D)=C}{\operatorname{argmin}} L(N, D). \qquad (1)$$

The functions $N_{opt}(C)$, and $D_{opt}(C)$ describe the optimal allocation of a computational budget $C$. We empirically estimate these functions based on the losses of over 400 models, ranging from under 70M to over 16B parameters, and trained on 5B to over 400B tokens – with each model configuration trained for several different training horizons. Our approach leads to considerably different results than that of [23]. We highlight our results in Figure 1 and how our approaches differ in Section 2.

Based on our estimated compute-optimal frontier, we predict that for the compute budget used to train *Gopher*, an optimal model should be 4 times smaller, while being training on 4 times more tokens. We verify this by training a more *compute-optimal* 70B model, called *Chinchilla*, on 1.4 trillion tokens. Not only does *Chinchilla* outperform its much larger counterpart, *Gopher*, but its reduced model size reduces inference cost considerably and greatly facilitates downstream uses on smaller hardware. The energy cost of a large language model is amortized through its usage for inference and fine-tuning. The benefits of a more optimally trained smaller model, therefore, extend beyond the immediate benefits of its improved performance.

## 2 Related Work

**Large language models.** A variety of large language models have been introduced in the last few years. These include both dense transformer models [6, 30, 48, 38, 52] and mixture-of-expert (MoE) models [11, 12, 60]. The largest dense transformers have passed 500 billion parameters [48, 8]. The drive to train larger and larger models is clear—so far increasing the size of language models has been responsible for improving the state-of-the-art in many language modelling tasks. Nonetheless, large language models face several challenges, including their overwhelming computational requirements (the cost of training and inference increase with model size) [38, 52] and the need for acquiring more high-quality training data. In fact, in this work we find that larger, high quality datasets will play a key role in any further scaling of language models. Concurrent to our work, a 540 billion parameter model trained on 768 billion tokens was released– PaLM [8]. While this model outperforms *Chinchilla*, it uses approximately $5\times$ the compute and is nearly $8\times$ larger, making it more difficult to use.

**Modelling the scaling behavior.** Understanding the scaling behaviour of language models and their transfer properties has been important in the development of recent large models [23, 18]. Kaplan et al. [23] first showed a predictable relationship between model size and loss over many orders of magnitude. The authors investigate the question of choosing the optimal model size to train for a given compute budget. Similar to us, they address this question by training various models. Our work differs from Kaplan et al. [23] in several important ways. First, the authors use a fixed number of training tokens and learning rate schedule for all models; this prevents them from modelling the impact of these hyperparameters on the loss. In contrast, we find that setting the learning rate schedule to approximately match the number of training tokens results in the best final loss regardless of model size—see Figure A1. For a fixed learning rate cosine schedule to 130B tokens, the intermediate loss estimates (for $D' << 130B$) are therefore overestimates of the loss of a model trained with a schedule length matching $D'$. Using these intermediate losses results in underestimating the effectiveness of training models on less data than 130B tokens, and eventually contributes to the conclusion that model size should increase faster than training data size as compute budget increases. In contrast, our analysis predicts that both quantities should scale at roughly the same rate. Secondly, we include models with up to 16B parameters, as we observe that there is slight curvature in the FLOP-loss frontier (see Appendix E)—in fact, the majority of the models used in our analysis have more than 500 million parameters, in contrast the majority of runs in [23] are significantly smaller—many being less than 100M parameters. Clark et al. [9] specifically looked in to the scaling properties of Mixture of Expert language models, showing that the scaling with number of experts diminishes as the model size increases—their approach models the loss as a function of two variables: the model size and the number of experts. However, the analysis is done with a fixed number of tokens, potentially underestimating the improvements of branching.

**Estimating hyperparameters for large models.** The model size and the number of training tokens are not the only two parameters to chose when selecting a language model and a procedure to train

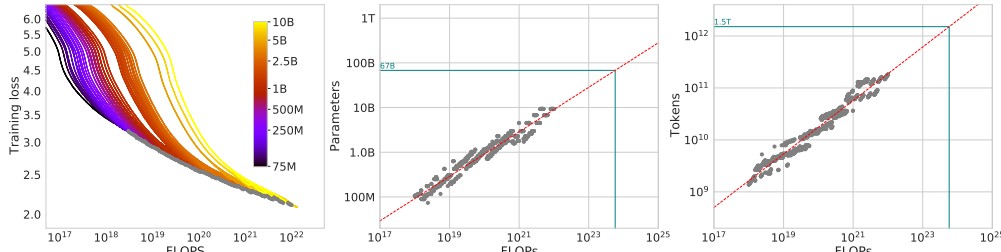

Figure 2: **Training curve envelope.** On the **left** we show all of our different runs. We launched a range of model sizes going from 70M to 10B, each for four different cosine cycle lengths. From these curves, we extracted the envelope of minimal loss per FLOP, and we used these points to estimate the optimal model size (**center**) for a given compute budget and the optimal number of training tokens (**right**). In green, we show projections of optimal model size and training token count based on the number of FLOPs used to train *Gopher* ($5.76 \times 10^{23}$).

it. Other important factors include learning rate, learning rate schedule, batch size, optimiser, and width-to-depth ratio. In this work, we focus on model size and the number of training steps, and we rely on existing work and provided experimental heuristics to determine the other necessary hyperparameters. Yang et al. [57] investigates how to choose a variety of these parameters for training an autoregressive transformer, including the learning rate and batch size. McCandlish et al. [33] finds only a weak dependence between optimal batch size and model size. Shallue et al. [46], Zhang et al. [59] suggest that using larger batch-sizes than those we use is possible. Levine et al. [28] investigates the optimal depth-to-width ratio for a variety of standard model sizes. We use slightly less deep models than proposed as this translates to better wall-clock performance on our hardware.

**Improved model architectures.** Recently, various promising alternatives to traditional dense transformers have been proposed. For example, through the use of conditional computation large MoE models like the 1.7 trillion parameter Switch transformer [12], the 1.2 Trillion parameter GLaM model [11], and others [1, 60] are able to provide a large effective model size despite using relatively fewer training and inference FLOPs. However, for very large models the computational benefits of routed models seems to diminish [9]. An orthogonal approach to improving language models is to augment transformers with explicit retrieval mechanisms, as done by [4, 15, 29]. This approach effectively increases the number of data tokens seen during training (by a factor of $\sim 10$ in [4]). This suggests that the performance of language models may be more dependant on the size of the training data than previously thought.

## 3   Estimating the optimal parameter/training tokens allocation

We present three different approaches to answer the question driving our research: *Given a fixed FLOPs budget, how should one trade-off model size and the number of training tokens?* In all three cases we start by training a range of models varying both model size and the number of training tokens and use the resulting training curves to fit an empirical estimator of how they should scale. We assume a power-law relationship between compute and model size as done in [9, 23], though future work may want to include potential curvature in this relationship for large model sizes. The resulting predictions are similar for all three methods and suggest that parameter count and number of training tokens should be increased equally with more compute —with proportions reported in Table 2. This is in clear contrast to previous work on this topic and warrants further investigation.

### 3.1   Approach 1: Fix model sizes and vary number of training tokens

In our first approach we vary the number of training steps for a fixed family of models (ranging from 70M to over 10B parameters), training each model for 4 different number of training sequences. From these runs, we are able to directly extract an estimate of the minimum loss achieved for a given number of training FLOPs. Training details for this approach can be found in Appendix D.

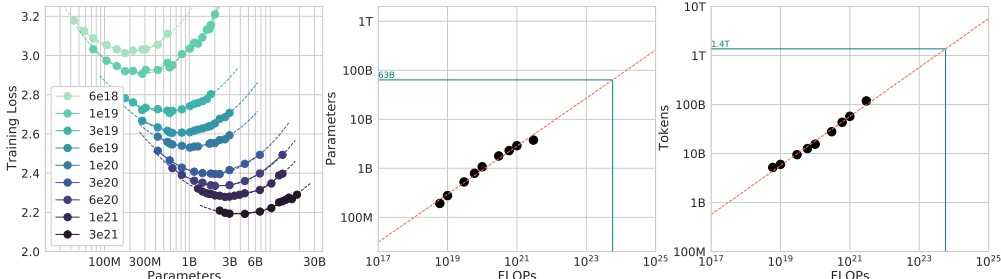

Figure 3: **IsoFLOP curves.** For various model sizes, we choose the number of training tokens such that the final FLOPs is a constant. The cosine cycle length is set to match the target FLOP count. We find a clear valley in loss, meaning that for a given FLOP budget there is an optimal model to train (**left**). Using the location of these valleys, we project optimal model size and number of tokens for larger models (**center** and **right**). In green, we show the estimated number of parameters and tokens for an *optimal* model trained with the compute budget of *Gopher*.

For each parameter count $N$ we train 4 different models: each uses a different horizon (measured in number of training tokens) over which we decay the learning rate by a factor of $10\times$; the range of horizons varies by a factor of $16\times$ for each parameter count. We smooth and linearly interpolate each training loss curve. From this, we obtain a continuous mapping from FLOP count to training loss for each run. We then determine which run achieves the lowest loss for each FLOP count. Using these interpolants, we obtain a mapping from FLOP count $C$ to the most efficient choice of model size $N_{opt}$ and number of training tokens $D_{opt}$ such that FLOPs$(N_{opt}, D_{opt}) = C$.[3] We apply this mapping onto logarithmically spaced values of $C$ and obtain many empirical triplets $(C_i, N_{opt,i}, D_{opt,i})_i$. Finally, we fit power laws to these empirical data, estimating $a$ and $b$ such that $N_{opt} \propto C^a$ and $D_{opt} \propto C^b$. We find that $a = 0.50$ and $b = 0.50$—as summarized in Table 2. We perform a simple experiment for early validating our analysis: given a budget of $10^{21}$ FLOPs, we compare the performance of training a model with a size recommended by our analysis to training a model with a size suggested by the analysis of [23]—using the model size we predict has a clear advantage (Section D.4).

### 3.2 Approach 2: IsoFLOP profiles

In our second approach we vary the model size for a fixed set of 9 different training FLOP counts (ranging from $6 \times 10^{18}$ to $3 \times 10^{21}$ FLOPs), and consider the final training loss for each point. This differs from the first approach that considers points $(N_i, D_i, L_i)_i$ along the entire training runs; the data points are here scarcer but more representative of the performance of a fully trained model.

For each FLOP budget, we plot the final loss (after smoothing) against the parameter count in Figure 3 (left). In all cases, we ensure that we have trained a diverse enough set of model sizes to see a clear minimum in the loss. We fit a parabola to each IsoFLOPs curve to directly estimate at what model size the minimum loss is achieved (Figure 3 (left)). As with the previous approach, we then fit a power law between FLOPs and loss-optimal model size and number of training tokens, shown in Figure 3 (center, right). Again, we fit exponents of the form $N_{opt} \propto C^a$ and $D_{opt} \propto C^b$ and we find that $a = 0.49$ and $b = 0.51$—as summarized in Table 2.

### 3.3 Approach 3: Fitting a parametric loss function

Lastly, we model all final losses from experiments in Approach 1 & 2 as a parametric function of model parameter count and the number of seen tokens. Following a classical risk decomposition (see Section D.2), we propose the following functional form

$$\hat{L}(N, D) \triangleq E + \frac{A}{N^\alpha} + \frac{B}{D^\beta}. \tag{2}$$

The first term captures the loss for an ideal generative process on the data distribution, and should correspond to the entropy of natural text. The second term captures the fact that a perfectly trained

---

[3]Note that all selected points are within the last 15% of training. Thus, when training a model over $D$ tokens, we should pick a cosine cycle length that decays $10\times$ over approximately $D$ tokens—see Appendix B.

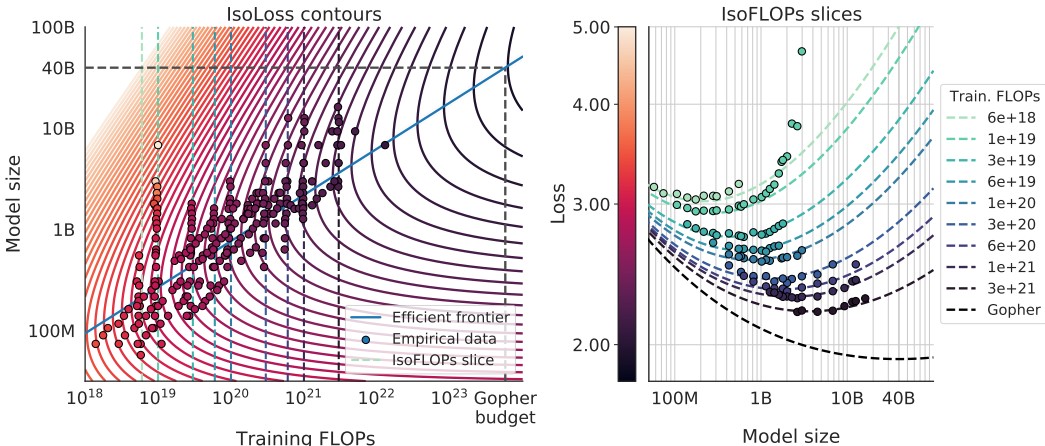

Figure 4: **Parametric fit.** We fit a parametric modelling of the loss $\hat{L}(N, D)$ and display contour (**left**) and isoFLOP slices (**right**). For each isoFLOP slice, we include a corresponding dashed line in the left plot. In the left plot, we show the efficient frontier in blue, which is a line in log-log space. Specifically, the curve goes through each iso-loss contour at the point with the fewest FLOPs. We project the optimal model size given the *Gopher* FLOP budget to be 40B parameters.

transformer with $N$ parameters underperforms the ideal generative process. The final term captures the fact that the transformer is not trained to convergence, as we only make a finite number of optimisation steps, on a sample of the dataset distribution.

**Model fitting.**    To estimate $(A, B, E, \alpha, \beta)$, we minimize the Huber loss [19] between the predicted and observed log loss using the L-BFGS algorithm [36]:

$$\min_{A,B,E,\alpha,\beta} \quad \sum_{\text{Runs } i} \text{Huber}_\delta \left( \log \hat{L}(N_i, D_i) - \log L_i \right) \tag{3}$$

We account for possible local minima by selecting the best fit from a grid of initialisations. The Huber loss ($\delta = 10^{-3}$) is robust to outliers, which we find important for good predictive performance over held-out data points. Section D.2 details the fitting procedure and the loss decomposition.

**Efficient frontier.**    We approximate the functions $N_{opt}$ and $D_{opt}$ by minimizing the parametric loss $\hat{L}$ under the constraint FLOPs$(N, D) \approx 6ND$ [23]. The resulting $N_{opt}$ and $D_{opt}$ balance the two terms in Equation (3) that depend on model size and data. By construction, they have a power-law form. We show contours of the fitted function $\hat{L}$ in Figure 4 (left), and the closed-form efficient computational frontier in blue. From this approach, we find that $a = 0.46$ and $b = 0.54$—as summarized in Table 2.

### 3.4   Optimal model scaling

We find that the three approaches, despite using different fitting methodologies and different trained models, yield comparable predictions for the optimal scaling in parameters and tokens with FLOPs (shown in Table 2). All three approaches suggest that as compute budget increases, model size and the amount of training data should be increased in approximately equal proportions. The first and second approaches yield very similar predictions for optimal model sizes, as shown in Figure 1 and Figure A3. The third approach predicts even smaller models being optimal at larger compute budgets. We note that the observed points $(L, N, D)$ for low training FLOPs ($C \leq 1e21$) have larger residuals $\|L - \hat{L}(N, D)\|_2^2$ than points with higher computational budgets. The fitted model places increased weight on the points with more FLOPs—automatically considering the low-computational budget points as outliers due to the Huber loss. As a consequence of the empirically observed negative curvature in the frontier $C \rightarrow N_{opt}$ (see Appendix E), this results in predicting a lower $N_{opt}$ than the two other approaches.

Table 2: **Estimated parameter and data scaling with increased training compute.** The listed values are the exponents, $a$ and $b$, on the relationship $N_{opt} \propto C^a$ and $D_{opt} \propto C^b$. Our analysis suggests a near equal scaling in parameters and data with increasing compute which is in clear contrast to previous work on the scaling of large models. The 10th and 90th percentiles are estimated via bootstrapping data (80% of the dataset is sampled 100 times) and are shown in parenthesis.

| Approach | Coeff. $a$ where $N_{opt} \propto C^a$ | Coeff. $b$ where $D_{opt} \propto C^b$ |
|---|---|---|
| 1. Minimum over training curves | 0.50 (0.488, 0.502) | 0.50 (0.501, 0.512) |
| 2. IsoFLOP profiles | 0.49 (0.462, 0.534) | 0.51 (0.483, 0.529) |
| 3. Parametric modelling of the loss | 0.46 (0.454, 0.455) | 0.54 (0.542, 0.543) |
| Kaplan *et al.* (2020) [23] | 0.73 | 0.27 |

In Table A3 we show the estimated number of FLOPs and tokens that would ensure that a model of a given size lies on the compute-optimal frontier. Our findings suggests that the current generation of large language models are considerably over-sized, given their respective compute budgets, as shown in Figure 1. Furthermore, the amount of training data that is projected to be needed is far beyond what is currently used to train large models, and underscores the importance of dataset collection in addition to engineering improvements that allow for model scale. While there is significant uncertainty extrapolating out many orders of magnitude, our analysis clearly suggests that given the training compute budget for many current LLMs, smaller models should have been trained on more tokens to achieve the most performant model. In Appendix C, we reproduce the IsoFLOP analysis on two additional datasets: C4 [40] and GitHub code [38]. In both cases we reach the similar conclusion that model size and number of training tokens should be scaled in equal proportions.

## 4    *Chinchilla*

Based on our analysis in Section 3, the optimal model size for the *Gopher* compute budget is somewhere between 40 and 70 billion parameters. We test this hypothesis by training a model on the larger end of this range—70B parameters—for 1.4T tokens, due to both dataset and computational efficiency considerations. In this section we compare this model, which we call *Chinchilla*, to *Gopher* and other LLMs. Both *Chinchilla* and *Gopher* have been trained for the same number of FLOPs but differ in the size of the model and the number of training tokens. While pre-training a large language model has a considerable compute cost, downstream fine-tuning and inference also make up substantial compute usage [38]. Due to being $4\times$ smaller than *Gopher*, both the memory footprint and inference cost of *Chinchilla* are also smaller.

### 4.1   Model and training details

The full set of hyperparameters used to train *Chinchilla* are given in Table 3. *Chinchilla* uses the same model architecture and training setup as *Gopher* with the exception of the differences listed below.

- We train *Chinchilla* on *MassiveText* (the same dataset as *Gopher*) but use a slightly different subset distribution (Table A1) to account for the increased number of training tokens.

- We use AdamW [32] for *Chinchilla* rather than Adam [24] as this improves the language modelling loss and the downstream task performance after finetuning.[4]

- We train *Chinchilla* with a slightly modified SentencePiece [25] tokenizer that does not apply NFKC normalisation. The vocabulary is very similar– 94.15% of tokens are the same as those used for training *Gopher*. We find that this particularly helps with the representation of mathematics and chemistry, for example.

- Whilst the forward and backward pass are computed in `bfloat16`, we store a `float32` copy of the weights in the distributed optimiser state [41]. See *Lessons Learned* from [38] for additional details.

---

[4]A model trained with AdamW only passes the training performance of a model trained with Adam around 80% of the way through the cosine cycle, though the ending performance is notably better– see Figure A7

In Appendix G we show the impact of the various optimiser related changes between *Chinchilla* and *Gopher*. All models in this analysis have been trained on TPUv3/TPUv4 [22] with JAX [5] and Haiku [17]. We include a *Chinchilla* model card [35] in Table A13.

## 4.2 Results

We perform an extensive evaluation of *Chinchilla*, comparing against various large language models. We evaluate on a large subset of the tasks presented in [38], shown in Table A6. As the focus of this work is on optimal model scaling, we included a large representative subset, and introduce a few new evaluations to allow for better comparison to other existing large models. The evaluation details for all tasks are the same as described in [38].

**Language modelling.** *Chinchilla* significantly outperforms *Gopher* on all evaluation subsets of The Pile [13], as shown in Figure A8. Compared to Jurassic-1 (178B) [30], *Chinchilla* is more performant on all but two subsets– `dm_mathematics` and `ubuntu_irc`– see Table A7 for a raw bits-per-byte comparison. On Wikitext103 [34], *Chinchilla* reaches 7.16 perplexity compared to 7.75 for *Gopher*.

**MMLU.** The Massive Multitask Language Understanding (MMLU) benchmark [16] consists of a range of exam-like questions on academic subjects. In Table A8, we report *Chinchilla*'s average 5-shot performance on MMLU (the full breakdown of results is shown in Table A9). On this benchmark, *Chinchilla* significantly outperforms *Gopher* despite being much smaller, with an average accuracy of 67.6% (improving upon *Gopher* by 7.6%). Remarkably, *Chinchilla* even outperforms the expert forecast for June 2023 of 63.4% accuracy (see Table A8) [50]. Furthermore, *Chinchilla* achieves greater than 90% accuracy on 4 different individual tasks– `high_school_gov_and_politics`, `international_law`, `sociology`, and `us_foreign_policy`. To our knowledge, no other model has achieved greater than 90% accuracy on a subset. In Figure A9, we show a comparison to *Gopher* broken down by task.

**Reading comprehension.** On the final word prediction dataset LAMBADA [37], *Chinchilla* achieves 77.4% accuracy, compared to 74.5% accuracy from *Gopher* and 76.6% from MT-NLG 530B (see Table 4). On RACE-h and RACE-m [27], *Chinchilla* greatly outperforms *Gopher*, improving accuracy by more than 10% in both cases—see Table 4.

**BIG-bench.** We analysed *Chinchilla* on the same set of BIG-bench tasks [49] reported in [38]. Similar to what we observed in MMLU, *Chinchilla* outperforms *Gopher* on the vast majority of tasks (see Figure A10). We find that *Chinchilla* improves the average performance by 10.7%, reaching an accuracy of 65.1% versus 54.4% for *Gopher*. Full accuracy results can be found in Table A10.

**Common sense.** We evaluate *Chinchilla* on various common sense benchmarks: PIQA [3], SIQA [45], Winogrande [44], HellaSwag [58], and BoolQ [10]. We find that *Chinchilla* outperforms both *Gopher* and GPT-3 on all tasks and outperforms MT-NLG 530B on all but one task—see Table 5. On TruthfulQA [31], *Chinchilla* reaches 43.6%, 58.5%, and 66.7% accuracy with 0-shot, 5-shot, and 10-shot respectively. In comparison, *Gopher* achieved only 29.5% 0-shot and 43.7% 10-shot accuracy. In stark contrast with the findings of [31], the large improvements (14.1% in 0-shot accuracy) achieved by Chinchilla suggest that better modelling of the pre-training data alone can lead to substantial improvements on this benchmark.

**Closed-book question answering.** Results on closed-book question answering benchmarks are reported in Table A11. On the Natural Questions dataset [26], *Chinchilla* achieves new closed-book SOTA accuracies: 31.5% 5-shot and 35.5% 64-shot, compared to 21% and 28% respectively, for *Gopher*. On TriviaQA [21] we show results for both the filtered (previously used in retrieval and open-

Table 3: *Chinchilla* **architecture details.** We list the number of layers, the key/value size, the bottleneck activation size $d_{model}$, the maximum learning rate, and the training batch size (# tokens). The feed-forward size is always set to $4 \times d_{model}$. Note that we double the batch size midway through training for both *Chinchilla* and *Gopher*.

| Model | Layers | Number Heads | Key/Value Size | $d_{model}$ | Max LR | Batch Size |
|-------|--------|--------------|----------------|-------------|--------|------------|
| *Gopher* 280B | 80 | 128 | 128 | 16,384 | $4 \times 10^{-5}$ | 3M → 6M |
| *Chinchilla* 70B | 80 | 64 | 128 | 8,192 | $1 \times 10^{-4}$ | 1.5M → 3M |

Table 4: **Reading comprehension.** On RACE-h and RACE-m [27], *Chinchilla* considerably improves performance over *Gopher*. Note that GPT-3 and MT-NLG 530B use a different prompt format than we do on RACE-h/m, so results are not comparable to *Gopher* and *Chinchilla*. On LAMBADA [37], *Chinchilla* outperforms both *Gopher* and MT-NLG 530B.

|  | *Chinchilla* | *Gopher* | GPT-3 | MT-NLG 530B |
|---|---|---|---|---|
| LAMBADA Zero-Shot | **77.4** | 74.5 | 76.2 | 76.6 |
| RACE-m Few-Shot | **86.8** | 75.1 | 58.1 | - |
| RACE-h Few-Shot | **82.3** | 71.6 | 46.8 | 47.9 |

Table 5: **Zero-shot comparison on Common Sense benchmarks.** We show a comparison between *Chinchilla*, *Gopher*, and MT-NLG 530B on various Common Sense benchmarks. We see that *Chinchilla* matches or outperforms *Gopher* and GPT-3 on all tasks. On all but one *Chinchilla* outperforms the much larger MT-NLG 530B model.

|  | *Chinchilla* | *Gopher* | GPT-3 | MT-NLG 530B | Supervised SOTA |
|---|---|---|---|---|---|
| HellaSWAG | **80.8%** | 79.2% | 78.9% | 80.2% | 93.9% |
| PIQA | 81.8% | 81.8% | 81.0% | **82.0%** | 90.1% |
| Winogrande | **74.9%** | 70.1% | 70.2% | 73.0% | 91.3% |
| SIQA | **51.3%** | 50.6% | - | - | 83.2% |
| BoolQ | **83.7**% | 79.3% | 60.5% | 78.2% | 91.4% |

book work) and unfiltered set (previously used in large language model evaluations). In both cases, *Chinchilla* substantially out performs *Gopher*. On the filtered version, Chinchilla lags behind the open book SOTA [20] by 7.9%. On the unfiltered set, *Chinchilla* outperforms GPT-3 (see Table A11).

## 5   Discussion & Conclusion

The trend so far in large language model training has been to increase the model size, often without increasing the number of training tokens. The largest dense transformer, MT-NLG 530B, is now over $3\times$ larger than GPT-3's 170 billion parameters from just two years ago. However, this model, as well as the majority of existing large models, have all been trained for a comparable number of tokens—around 300 billion. While the desire to train these mega-models has led to substantial engineering innovation, we hypothesize that the race to train larger and larger models is resulting in models that are substantially underperforming compared to what could be achieved with the same compute budget.

We propose three predictive approaches towards optimally setting model size and training duration, based on the outcome of over 400 training runs. All three approaches predict that *Gopher* is substantially over-sized and estimate that for the same compute budget a smaller model trained on more data will perform better. We directly test this hypothesis by training *Chinchilla*, a 70B parameter model, and show that it outperforms *Gopher* and even larger models on nearly every measured evaluation task.

Whilst our method allows us to make predictions on how to scale large models when given additional compute, there are several limitations. Due to the cost of training large models, we only have two comparable training runs at large scale (*Chinchilla* and *Gopher*), and we do not have additional tests at intermediate scales. Furthermore, we assume that the efficient computational frontier can be described by a power-law relationship between the compute budget, model size, and number of training tokens. However, we observe some concavity in $\log(N_{opt})$ at high compute budgets (see Appendix E). This suggests that we may still be overestimating the optimal size of large models. Finally, the training runs for our analysis have all been trained on less than an epoch of data; future work may consider the multiple epoch regime. Despite these limitations, the comparison of *Chinchilla* to *Gopher* validates our performance predictions, that have thus enabled training a better (and more lightweight) model at the same compute budget.

Though there has been significant recent work allowing larger and larger models to be trained, our analysis suggests an increased focus on dataset scaling is needed. Speculatively, we expect that scaling to larger and larger datasets is only beneficial when the data is high-quality. This calls for responsibly collecting larger datasets with a high focus on dataset quality. Larger datasets will require extra care to ensure train-test set overlap is properly accounted for, both in the language modelling loss but also with downstream tasks. Finally, training for trillions of tokens introduces many ethical and privacy concerns. Large datasets scraped from the web will contain toxic language, biases, and private information. With even larger datasets being used, the quantity (if not the frequency) of such information increases, which makes dataset introspection all the more important. *Chinchilla* does suffer from bias and toxicity but interestingly it seems less affected than *Gopher* (see Appendix I).. Better understanding how performance of large language models and toxicity interact is an important future research question.

While we have applied our methodology towards the training of auto-regressive language models, we expect that there is a similar trade-off between model size and the amount of data in other modalities. As training large models is very expensive, choosing the optimal model size and training steps beforehand is essential. The methods we propose are easy to reproduce in new settings.

## Acknowledgments and Disclosure of Funding

We'd like to thank Jean-baptiste Alayrac, Kareem Ayoub, Chris Dyer, Nando de Freitas, Demis Hassabis, Geoffrey Irving, Koray Kavukcuoglu, Nate Kushman and Angeliki Lazaridou for useful comments on the manuscript. We'd like to thank Andy Brock, Irina Higgins, Michela Paganini, Francis Song, and other colleagues at DeepMind for helpful discussions. We are also very grateful to the JAX and XLA team for their support and assistance.

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
