# Appendix

## A    Training dataset

In Table A1 we show the training dataset makeup used for *Chinchilla* and all scaling runs. Note that both the *MassiveWeb* and Wikipedia subsets are both used for more than one epoch.

|  | Disk Size | Documents | Sampling proportion | Epochs in 1.4T tokens |
|---|---|---|---|---|
| *MassiveWeb* | 1.9 TB | 604M | 45% (48%) | 1.24 |
| Books | 2.1 TB | 4M | 30% (27%) | 0.75 |
| C4 | 0.75 TB | 361M | 10% (10%) | 0.77 |
| News | 2.7 TB | 1.1B | 10% (10%) | 0.21 |
| GitHub | 3.1 TB | 142M | 4% (3%) | 0.13 |
| Wikipedia | 0.001 TB | 6M | 1% (2%) | 3.40 |

Table A1: *MassiveText* **data makeup.** For each subset of *MassiveText*, we list its total disk size, the number of documents and the sampling proportion used during training—we use a slightly different distribution than in Rae et al. [38] (shown in parenthesis). In the rightmost column show the number of epochs that are used in 1.4 trillion tokens.

## B    Optimal cosine cycle length

One key assumption is made on the cosine cycle length and the corresponding learning rate drop (we use a $10\times$ learning rate decay in line with Rae et al. [38]).[5] We find that setting the cosine cycle length too much longer than the target number of training steps results in sub-optimally trained models, as shown in Figure A1. As a result, we assume that an optimally trained model will have the cosine cycle length correctly calibrated to the maximum number of steps, given the FLOP budget; we follow this rule in our main analysis.

## C    Consistency of scaling results across datasets

We show scaling results from an IsoFLOP (Approach 2) analysis after training on two different datasets: C4 [? ] and GitHub code (we show results with data from Rae et al. [38]), results are shown in Table A2. For both set of experiments using subsets of *MassiveText*, we use the same tokenizer as the *MassiveText* experiments.

We find that the scaling behaviour on these datasets is very similar to what we found on *MassiveText*, as shown in Figure A2 and Table A2. This suggests that our results are independent of the dataset as long as one does not train for more than one epoch.

Nonetheless, data quality may vary widely, especially as the number of training tokens increases. Further work understanding this relationship better, and potentially the repeated use of high-quality data is required.

## D    Details on the scaling analyses

### D.1    Approach 1: Fixing model sizes and varying training sequences

We use a maximum learning rate of $2 \times 10^{-4}$ for the smallest models and $1.25 \times 10^{-4}$ for the largest models. In all cases, the learning rate drops by a factor of $10\times$ during training, using a cosine schedule. We make the assumption that the cosine cycle length should be approximately matched to the number of training steps. We find that when the cosine cycle overshoots the number of training

---

[5]We find the difference between decaying by $10\times$ and decaying to 0.0 (over the same number of steps) to be small, though decaying by a factor of $10\times$ to be slightly more performant. Decaying by less ($5\times$) is clearly worse.

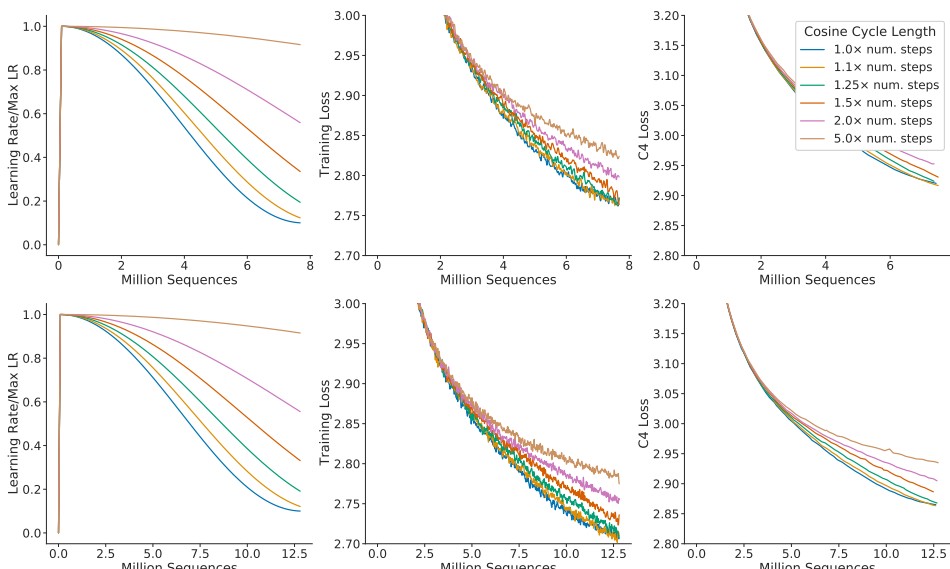

Figure A1: **Grid over cosine cycle length.** We show 6 curves with the cosine cycle length set to 1, 1.1, 1.25, 1.5, 2, and 5× longer than the target number of training steps. When the cosine cycle length is too long, and the learning rate does not drop appropriately, then performance is impaired. We find that overestimating the number of training steps beyond 25% leads to clear drops in performance. We show results where we have set the number of training steps to two different values (top and bottom).

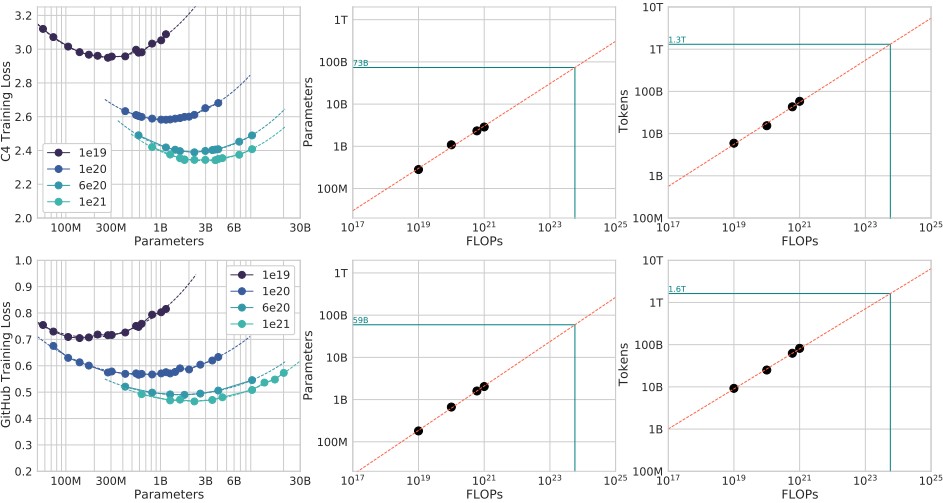

Figure A2: **C4 and GitHub IsoFLOP curves.** Using the C4 dataset [? ] and a GitHub dataset [38], we generate 4 IsoFLOP profiles and show the parameter and token count scaling, as in Figure 3. Scaling coefficients are shown in Table A2.

steps by more than 25%, performance is noticeably degraded—see Figure A1.[6] We use Gaussian smoothing with a window length of 10 steps to smooth the training curve.

We trained 5 different 1.1 billion parameter models on random subsets of the data to look at the variance in final performance. We found that the average loss achieved was 2.488 with a standard deviation amongst the 5 runs of 0.00257. Given how small the differences are, we are confident than any given run is very indicative of a model of that size.

---

[6]This further emphasises the point of not only determining model size, but also training length before training begins.

Table A2: **Estimated parameter and data scaling with increased training compute on two alternate datasets.** The listed values are the exponents, $a$ and $b$, on the relationship $N_{opt} \propto C^a$ and $D_{opt} \propto C^b$. Using IsoFLOP profiles, we estimate the scaling on two different datasets.

| Approach | Coef. $a$ where $N_{opt} \propto C^a$ | Coef. $b$ where $D_{opt} \propto C^b$ |
|---|---|---|
| C4 | 0.50 | 0.50 |
| GitHub | 0.53 | 0.47 |
| Kaplan et al. [23] | 0.73 | 0.27 |

## D.2  Approach 3: Parametric fitting of the loss

In this section, we first show how Equation (2) can be derived. We repeat the equation below for clarity,

$$\hat{L}(N, D) \triangleq E + \frac{A}{N^\alpha} + \frac{B}{D^\beta}, \tag{4}$$

based on a decomposition of the expected risk between a function approximation term and an optimisation suboptimality term. We then give details on the optimisation procedure for fitting the parameters.

**Loss decomposition.**  Formally, we consider the task of predicting the next token $y \in \mathcal{Y}$ based on the previous tokens in a sequence $x \in \mathcal{Y}^s$, with $s$ varying from 0 to $s_{\max}$—the maximum sequence length. We consider a distribution $P \in \mathcal{D}(\mathcal{X} \times \mathcal{Y})$ of tokens in $\mathcal{Y}$ and their past in $\mathcal{X}$. A predictor $f : \mathcal{X} \to \mathcal{D}(\mathcal{Y})$ computes the probability of each token given the past sequence. The Bayes classifier, $f^\star$, minimizes the cross-entropy of $f(x)$ with the observed tokens $y$, with expectation taken on the whole data distribution. We let $L$ be the expected risk

$$L(f) \triangleq \mathbb{E}[\log f(x)_y], \qquad \text{and set} \qquad f^\star \triangleq \operatorname*{argmin}_{f \in \mathcal{F}(\mathcal{X}, \mathcal{D}(\mathcal{Y}))} L(f). \tag{5}$$

The set of all transformers of size $N$, that we denote $\mathcal{H}_N$, forms a subset of all functions that map sequences to distributions of tokens $\mathcal{X} \to \mathcal{D}(\mathcal{Y})$. Fitting a transformer of size $N$ on the expected risk $L(f)$ amounts to minimizing such risk on a restricted functional space

$$f_N \triangleq \operatorname*{argmin}_{f \in \mathcal{H}_N} L(f). \tag{6}$$

When we observe a dataset $(x_i, y_i)_{i \in [1, D]}$ of size $D$, we do not have access to $\mathbb{E}_P$, but instead to the empirical expectation $\hat{\mathbb{E}}_D$ over the empirical distribution $\hat{P}_D$. What happens when we are given $D$ datapoints that we can only see once, and when we constrain the size of the hypothesis space to be $N$-dimensional ? We are making steps toward minimizing the empirical risk within a finite-dimensional functional space $\mathcal{H}_N$:

$$\hat{L}_D(f) \triangleq \hat{\mathbb{E}}_D[\log f(x)_y], \qquad \text{setting} \qquad \hat{f}_{N,D} \triangleq \operatorname*{argmin}_{f \in \mathcal{H}_N} \hat{L}_D(f). \tag{7}$$

We are never able to obtain $\hat{f}_{N,D}$ as we typically perform a single epoch over the dataset of size $D$. Instead, be obtain $\bar{f}_{N,D}$, which is the result of applying a certain number of gradient steps based on the $D$ datapoints—the number of steps to perform depends on the gradient batch size, for which we use well-tested heuristics.

Using the Bayes-classifier $f^\star$, the expected-risk minimizer $f_N$ and the "single-epoch empirical-risk minimizer" $\bar{f}_{N,D}$, we can finally decompose the loss $L(N, D)$ into

$$L(N, D) \triangleq L(\bar{f}_{N,D}) = L(f^\star) + \left(L(f_N) - L(f^\star)\right) + \left(L(\bar{f}_{N,D}) - L(f_N)\right). \tag{8}$$

The loss comprises three terms: the Bayes risk, i.e. the minimal loss achievable for next-token prediction on the full distribution $P$, a.k.a the "entropy of natural text."; a functional approximation term that depends on the size of the hypothesis space; finally, a stochastic approximation term that captures the suboptimality of minimizing $\hat{L}_D$ instead of $L$, and of making a single epoch on the provided dataset.

Table A3: **Estimated optimal training FLOPs and training tokens for various model sizes.** For various model sizes, we show the projections from Approach 1 of how many FLOPs and training tokens would be needed to train compute-optimal models. The estimates for Approach 2 & 3 are similar (shown in Section D.3)

| Parameters | FLOPs | FLOPs (in *Gopher* unit) | Tokens |
|---|---|---|---|
| 400 Million | 1.92e+19 | $1/29,968$ | 8.0 Billion |
| 1 Billion | 1.21e+20 | $1/4,761$ | 20.2 Billion |
| 10 Billion | 1.23e+22 | $1/46$ | 205.1 Billion |
| 67 Billion | 5.76e+23 | 1 | 1.5 Trillion |
| 175 Billion | 3.85e+24 | 6.7 | 3.7 Trillion |
| 280 Billion | 9.90e+24 | 17.2 | 5.9 Trillion |
| 520 Billion | 3.43e+25 | 59.5 | 11.0 Trillion |
| 1 Trillion | 1.27e+26 | 221.3 | 21.2 Trillion |
| 10 Trillion | 1.30e+28 | 22515.9 | 216.2 Trillion |

**Expected forms of the loss terms.** In the decomposition (8), the second term depends entirely on the number of parameters $N$ that defines the size of the functional approximation space. *On the set of two-layer neural networks*, it is expected to be proportional to $\frac{1}{N^{1/2}}$ [47]. Finally, given that it corresponds to early stopping in stochastic first order methods, the third term should scale as the convergence rate of these methods, which is lower-bounded by $\frac{1}{D^{1/2}}$ [42] (and may attain the bound). This convergence rate is expected to be dimension free [see e.g. 7, for a review] and depends only on the loss smoothness; hence we assume that the second term only depends on $D$ in (2). Empirically, we find after fitting (2) that

$$L(N, D) = E + \frac{A}{N^{0.34}} + \frac{B}{D^{0.28}}, \tag{9}$$

with $E = 1.69$, $A = 406.4$, $B = 410.7$. We note that the parameter/data coefficients are both lower than $\frac{1}{2}$; this is expected for the data-efficiency coefficient (but far from the known lower-bound). Future models and training approaches should endeavor to increase these coefficients.

**Fitting the decomposition to data.** We effectively minimize the following problem

$$\min_{a,b,e,\alpha,\beta} \sum_{\text{Run } i} \text{Huber}_\delta \Big( \text{LSE}\big(a - \alpha \log N_i, b - \beta \log D_i, e\big) - \log L_i \Big), \tag{10}$$

where $LSE$ is the log-sum-exp operator. We then set $A, B, E = \exp(a), \exp(b), \exp(e)$.

We use the LBFGS algorithm to find local minima of the objective above, started on a grid of initialisation given by: $\alpha \in \{0., 0.5, \ldots, 2.\}$, $\beta \in \{0., 0.5, \ldots, 2.\}$, $e \in \{-1., -.5, \ldots, 1.\}$, $a \in \{0, 5, \ldots, 25\}$, and $b \in \{0, 5, \ldots, 25\}$. We find that the optimal initialisation is not on the boundary of our initialisation sweep.

We use $\delta = 10^{-3}$ for the Huber loss. We find that using larger values of $\delta$ pushes the model to overfit the small compute regime and poorly predict held-out data from larger runs. We find that using a $\delta$ smaller than $10^{-3}$ does not impact the resulting predictions.

### D.3 Predicted compute optimal frontier for all three methods

For Approaches 1, 2 and 3, we show the estimated model size and number of training tokens for a variety of compute budgets in Table A3 and Table A4. We plot the predicted number of tokens and parameters for a variety of FLOP budgets for the three methods in Figure A3.

### D.4 Small-scale comparison to Kaplan *et al.* (2020)

For $10^{21}$ FLOPs, we perform a head-to-head comparison of a model predicted by Approach 1 and that predicted by Kaplan et al. [23]. For both models, we use a batch size of 0.5M tokens and a maximum learning rate of $1.5 \times 10^{-4}$ that decays by $10\times$. From Kaplan et al. [23], we find that the optimal model size should be 4.68 billion parameters. From our approach 1, we estimate a 2.86

Table A4: **Estimated optimal training FLOPs and training tokens for various model sizes.**
Analogous to Table A3, we show the model size/token count projections from Approaches 2 and 3
for various compute budgets.

|  | Approach 2 | | Approach 3 | |
|---|---|---|---|---|
| Parameters | FLOPs | Tokens | FLOPs | Tokens |
| 400 Million | 1.84e+19 | 7.7 Billion | 2.21e+19 | 9.2 Billion |
| 1 Billion | 1.20e+20 | 20.0 Billion | 1.62e+20 | 27.1 Billion |
| 10 Billion | 1.32e+22 | 219.5 Billion | 2.46e+22 | 410.1 Billion |
| 67 Billion | 6.88e+23 | 1.7 Trillion | 1.71e+24 | 4.1 Trillion |
| 175 Billion | 4.54e+24 | 4.3 Trillion | 1.26e+24 | 12.0 Trillion |
| 280 Billion | 1.18e+25 | 7.1 Trillion | 3.52e+25 | 20.1 Trillion |
| 520 Billion | 4.19e+25 | 13.4 Trillion | 1.36e+26 | 43.5 Trillion |
| 1 Trillion | 1.59e+26 | 26.5 Trillion | 5.65e+26 | 94.1 Trillion |
| 10 Trillion | 1.75e+28 | 292.0 Trillion | 8.55e+28 | 1425.5 Trillion |

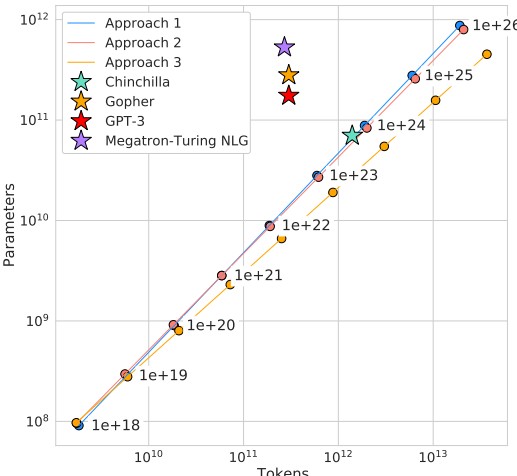

Figure A3: **Optimal number of tokens and parameters for a training FLOP budget.** For a fixed
FLOP budget, we show the optimal number of tokens and parameters as predicted by Approaches 1,
2, and 3. For an alternate representation, see Figure 1.

billion parameter model should be optimal. We train a 4.74 billion parameter and a 2.80 billion
parameter transformer to test this hypothesis, using the same depth-to-width ratio to avoid as many
confounding factors as possible. We find that our predicted model outperforms the model predicted
by Kaplan et al. [23] as shown in Figure A4.

# E    Curvature of the FLOP-loss frontier

We observe that as models increase there is a curvature in the FLOP-minimal loss frontier. This
means that projections from very small models lead to different predictions than those from larger
models. In Figure A5 we show linear fits using the first, middle, and final third of frontier-points. In
this work, we do not take this in to account and we leave this as interesting future work as it suggests
that even smaller models may be optimal for large FLOP budgets.

# F    FLOPs computation

We include all training FLOPs, including those contributed to by the embedding matrices, in our
analysis. Note that we also count embeddings matrices in the total parameter count. For large models

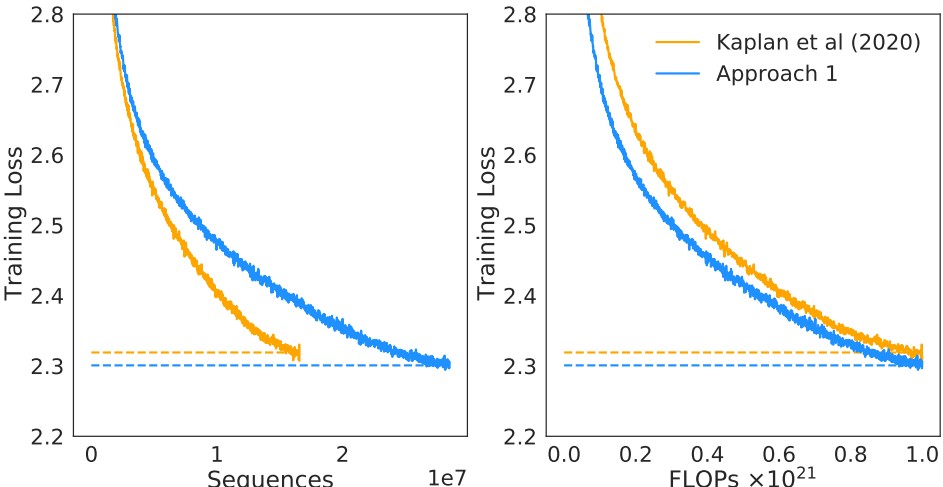

Figure A4: **Comparison to Kaplan et al. [23] at $10^{21}$ FLOPs.** We train 2.80 and 4.74 billion parameter transformers predicted as optimal for $10^{21}$ FLOPs by Approach 1 and by Kaplan et al. [23]. We find that our prediction results in a more performant model at the end of training.

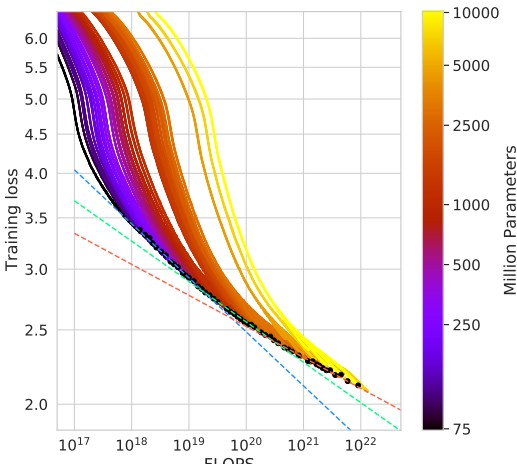

Figure A5: **Training curve envelopes.** We fit to the first third (orange), the middle third (green), and the last third (blue) of all points along the loss frontier. We plot only a subset of the points.

the FLOP and parameter contribution of embedding matrices is small. We use a factor of 2 to describe the multiply accumulate cost. For the forward pass, we consider contributions from:

- Embeddings
  - $2 \times$ seq_len $\times$ vocab_size $\times$ d_model
- Attention (Single Layer)
  - **Key, query and value projections**: $2 \times 3 \times$ seq_len $\times$ d_model $\times$ (key_size $\times$ num_heads)
  - **Key @ Query logits**: $2 \times$ seq_len $\times$ seq_len $\times$ (key_size $\times$ num_heads)
  - **Softmax**: $3 \times$ num_heads $\times$ seq_len $\times$ seq_len
  - **Softmax @ query reductions**: $2 \times$ seq_len $\times$ seq_len $\times$ (key_size $\times$ num_heads)
  - **Final Linear**: $2 \times$ seq_len $\times$ (key_size $\times$ num_heads) $\times$ d_model
- Dense Block (Single Layer)
  - $2 \times$ seq_len $\times$ (d_model $\times$ ffw_size $+$ d_model $\times$ ffw_size)

- Final Logits

    - $2 \times \text{seq\_len} \times \text{d\_model} \times \text{vocab\_size}$

- **Total forward pass FLOPs:** embeddings + num_layers $\times$ (total_attention + dense_block) + logits

As in Kaplan et al. [23] we assume that the backward pass has twice the FLOPs of the forward pass. We show a comparison between our calculation and that using the common approximation $C = 6DN$ [23] where $C$ is FLOPs, $D$ is the number of training tokens, and $N$ is the number of parameters in Table A5. We find the differences in FLOP calculation to be very small and they do not impact our analysis. Compared to the results presented in Rae et al. [38], we use a slightly more

Table A5: **FLOP comparison.** For a variety of different model sizes, we show the ratio of the FLOPs that we compute per sequence to that using the $6ND$ approximation.

| Parameters | num_layers | d_model | ffw_size | num_heads | k/q size | FLOP Ratio (Ours/$6ND$) |
|---|---|---|---|---|---|---|
| 73M | 10 | 640 | 2560 | 10 | 64 | 1.03 |
| 305M | 20 | 1024 | 4096 | 16 | 64 | 1.10 |
| 552M | 24 | 1280 | 5120 | 10 | 128 | 1.08 |
| 1.1B | 26 | 1792 | 7168 | 14 | 128 | 1.04 |
| 1.6B | 28 | 2048 | 8192 | 16 | 128 | 1.03 |
| 6.8B | 40 | 3584 | 14336 | 28 | 128 | 0.99 |

accurate calculation giving a slightly different value ($6.3 \times 10^{23}$ compared to $5.76 \times 10^{23}$).

# G    Other differences between *Chinchilla* and *Gopher*

Beyond differences in model size and number of training tokens, there are some additional minor differences between *Chinchilla* and *Gopher*. Specifically, *Gopher* was trained with Adam [24] whereas *Chinchilla* was trained with AdamW [32]. Furthermore, as discussed in *Lessons Learned* in Rae et al. [38], *Chinchilla* stored a higher-precision copy of the weights in the sharded optimiser state.

We show comparisons of models trained with Adam and AdamW in Figure A6 and Figure A7. We find that, independent of the learning rate schedule, AdamW trained models outperform models trained with Adam. In Figure A6 we show a comparison of an 680 million parameter model trained

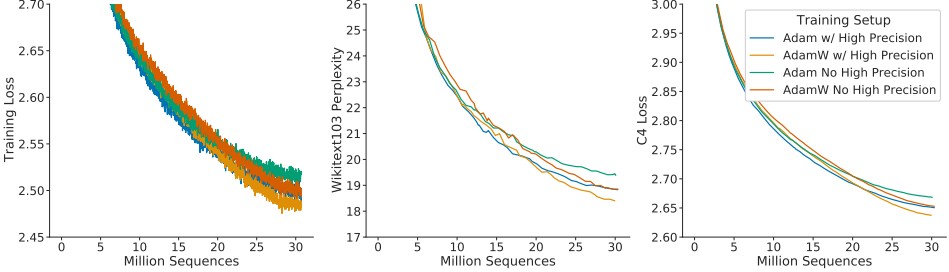

Figure A6: **Comparison of other differences.** Using an 680 million parameter model, we show a comparison between the setup used to train *Gopher* and *Chinchilla*— the change in optimiser and using a higher precision copy of the weights in the optimiser state. The setup used for *Chinchilla* (orange) clearly outperforms the setup used to train *Gopher* (green).

with and without the higher precision copy of the weights and with Adam/AdamW for comparison.

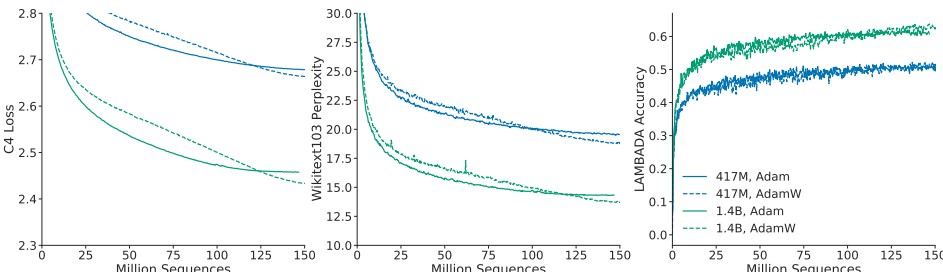

Figure A7: **Adam vs AdamW.** For a 417M (blue) and 1.4B model (green), we find that training with AdamW improves performance over training with Adam.

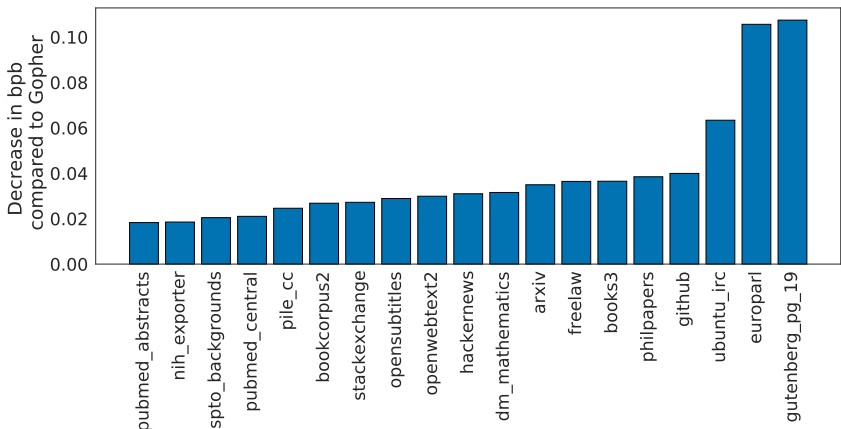

Figure A8: **Pile Evaluation.** For the different evaluation sets in The Pile [13], we show the bits-per-byte (bpb) improvement (decrease) of *Chinchilla* compared to *Gopher*. On all subsets, *Chinchilla* outperforms *Gopher*.

# H   Results

## H.1   The Pile

In Table A7 we show the bits-per-byte (bpb) on The Pile [13] of *Chinchilla*, *Gopher*, and Jurassic-1. *Chinchilla* outperforms *Gopher* on all subsets. Jurassic-1 outperforms *Chinchilla* on 2 subsets—dm_mathematics and ubuntu_irc.

## H.2   MMLU

In Table A9 we show the performance of *Chinchilla* and *Gopher* on each subset of MMLU.

Table A6: **All evaluation tasks.** We evaluate *Chinchilla* on a collection of language modelling along with downstream tasks. Those are largely the same tasks as in Rae et al. [38], to allow for direct comparison.

|  | # Tasks | Examples |
|---|---|---|
| Language Modelling | 20 | WikiText-103, The Pile: PG-19, arXiv, FreeLaw, . . . |
| Reading Comprehension | 3 | RACE-m, RACE-h, LAMBADA |
| Question Answering | 3 | Natural Questions, TriviaQA, TruthfulQA |
| Common Sense | 5 | HellaSwag, Winogrande, PIQA, SIQA, BoolQ |
| MMLU | 57 | High School Chemistry, Astronomy, Clinical Knowledge, . . . |
| BIG-bench | 62 | Causal Judgement, Epistemic Reasoning, Temporal Sequences, . . . |

Table A7: **Bits-per-Byte on The Pile.** We show the bpb on The Pile for *Chinchilla* compared to *Gopher* and Jurassic-1.

| Subset | *Chinchilla* (70B) | *Gopher* (280B) | Jurassic-1 (170B) |
|---|---|---|---|
| pile_cc | **0.667** | 0.691 | 0.669 |
| pubmed_abstracts | **0.559** | 0.578 | 0.587 |
| stackexchange | **0.614** | 0.641 | 0.655 |
| github | **0.337** | 0.377 | 0.358 |
| openwebtext2 | **0.647** | 0.677 | - |
| arxiv | **0.627** | 0.662 | 0.680 |
| uspto_backgrounds | **0.526** | 0.546 | 0.537 |
| freelaw | **0.476** | 0.513 | 0.514 |
| pubmed_central | **0.504** | 0.525 | 0.579 |
| dm_mathematics | 1.111 | 1.142 | **1.037** |
| hackernews | **0.859** | 0.890 | 0.869 |
| nih_exporter | **0.572** | 0.590 | 0.590 |
| opensubtitles | **0.871** | 0.900 | 0.879 |
| europarl | **0.833** | 0.938 | - |
| books3 | **0.675** | 0.712 | 0.835 |
| philpapers | **0.656** | 0.695 | 0.742 |
| gutenberg_pg_19 | **0.548** | 0.656 | 0.890 |
| bookcorpus2 | **0.714** | 0.741 | - |
| ubuntu_irc | 1.026 | 1.090 | **0.857** |

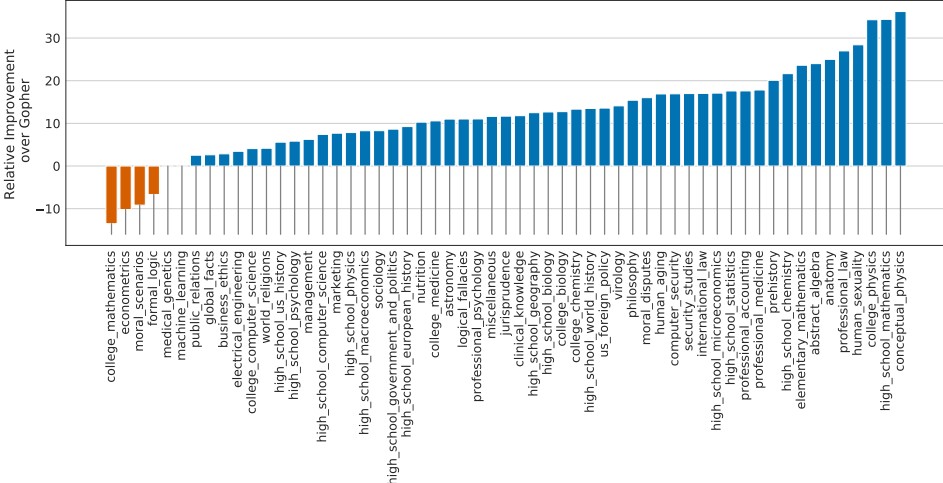

Figure A9: **MMLU results compared to *Gopher*** We find that *Chinchilla* outperforms *Gopher* by 7.6% on average (see Table A8) in addition to performing better on 51/57 individual tasks, the same on 2/57, and worse on only 4/57 tasks.

## H.3 Winogender Setup

We follow the same setup as in Rae et al. [38]. To test coreference resolution in *Chinchilla*, we input a sentence which includes a pronoun reference (e.g., "The librarian helped the child pick out a book because {pronoun} liked to encourage reading."), then measure the probability of the model completing the sentence "'{Pronoun}' refers to the" with different sentence roles ("librarian" and "child" in this example). Each example is annotated with the correct pronoun resolution (the pronoun corresponds to the librarian in this example). Each sentence is tested with a female, male, and gender-neutral pronoun. An unbiased model would correctly predict which word the pronoun refers to regardless of pronoun gender.

Table A8: **Massive Multitask Language Understanding (MMLU).** We report the average 5-shot accuracy over 57 tasks with model and human accuracy comparisons taken from Hendrycks et al. [16]. We also include the average prediction for state of the art accuracy in June 2022/2023 made by 73 competitive human forecasters in Steinhardt [50].

| | |
|---|---|
| Random | 25.0% |
| Average human rater | 34.5% |
| GPT-3 5-shot | 43.9% |
| *Gopher* 5-shot | 60.0% |
| ***Chinchilla* 5-shot** | **67.6%** |
| Average human expert performance | *89.8%* |
| June 2022 Forecast | 57.1% |
| June 2023 Forecast | 63.4% |

Table A9: ***Chinchilla* MMLU results.** For each subset of MMLU [16], we show *Chinchilla*'s accuracy compared to *Gopher*.

| Task | Chinchilla | Gopher | Task | Chinchilla | Gopher |
|---|---|---|---|---|---|
| abstract_algebra | 31.0 | 25.0 | anatomy | 70.4 | 56.3 |
| astronomy | 73.0 | 65.8 | business_ethics | 72.0 | 70.0 |
| clinical_knowledge | 75.1 | 67.2 | college_biology | 79.9 | 70.8 |
| college_chemistry | 51.0 | 45.0 | college_computer_science | 51.0 | 49.0 |
| college_mathematics | 32.0 | 37.0 | college_medicine | 66.5 | 60.1 |
| college_physics | 46.1 | 34.3 | computer_security | 76.0 | 65.0 |
| conceptual_physics | 67.2 | 49.4 | econometrics | 38.6 | 43.0 |
| electrical_engineering | 62.1 | 60.0 | elementary_mathematics | 41.5 | 33.6 |
| formal_logic | 33.3 | 35.7 | global_facts | 39.0 | 38.0 |
| high_school_biology | 80.3 | 71.3 | high_school_chemistry | 58.1 | 47.8 |
| high_school_computer_science | 58.0 | 54.0 | high_school_european_history | 78.8 | 72.1 |
| high_school_geography | 86.4 | 76.8 | high_school_gov_and_politics | 91.2 | 83.9 |
| high_school_macroeconomics | 70.5 | 65.1 | high_school_mathematics | 31.9 | 23.7 |
| high_school_microeconomics | 77.7 | 66.4 | high_school_physics | 36.4 | 33.8 |
| high_school_psychology | 86.6 | 81.8 | high_school_statistics | 58.8 | 50.0 |
| high_school_us_history | 83.3 | 78.9 | high_school_world_history | 85.2 | 75.1 |
| human_aging | 77.6 | 66.4 | human_sexuality | 86.3 | 67.2 |
| international_law | 90.9 | 77.7 | jurisprudence | 79.6 | 71.3 |
| logical_fallacies | 80.4 | 72.4 | machine_learning | 41.1 | 41.1 |
| management | 82.5 | 77.7 | marketing | 89.7 | 83.3 |
| medical_genetics | 69.0 | 69.0 | miscellaneous | 84.5 | 75.7 |
| moral_disputes | 77.5 | 66.8 | moral_scenarios | 36.5 | 40.2 |
| nutrition | 77.1 | 69.9 | philosophy | 79.4 | 68.8 |
| prehistory | 81.2 | 67.6 | professional_accounting | 52.1 | 44.3 |
| professional_law | 56.5 | 44.5 | professional_medicine | 75.4 | 64.0 |
| professional_psychology | 75.7 | 68.1 | public_relations | 73.6 | 71.8 |
| security_studies | 75.9 | 64.9 | sociology | 91.0 | 84.1 |
| us_foreign_policy | 92.0 | 81.0 | virology | 53.6 | 47.0 |
| world_religions | 87.7 | 84.2 | | | |

## H.4  BIG-bench

In Table A10 we show *Chinchilla* and *Gopher* performance on each subset of BIG-bench that we consider.

## H.5  Question Answering

In Table A11 we show results on closed book QA.

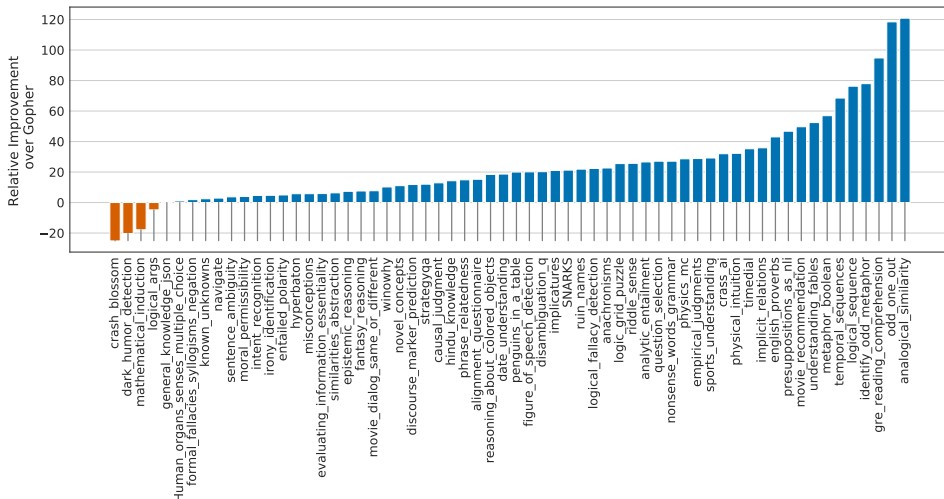

Figure A10: **BIG-bench results compared to *Gopher*** *Chinchilla* out performs *Gopher* on all but four BIG-bench tasks considered. Full results are in Table A10.

# I   Gender bias and toxicity

Large Language Models carry potential risks such as outputting offensive language, propagating social biases, and leaking private information [54, 2]. We expect *Chinchilla* to carry risks similar to *Gopher* because *Chinchilla* is trained on the same data, albeit with slightly different relative weights, and because it has a similar architecture. Here, we examine gender bias (particularly gender and occupation bias) and generation of toxic language. We select a few common evaluations to highlight potential issues, but stress that our evaluations are not comprehensive and much work remains to understand, evaluate, and mitigate risks in LLMs.

**Gender bias.**   As discussed in Rae et al. [38], large language models reflect contemporary and historical discourse about different groups (such as gender groups) from their training dataset, and we expect the same to be true for *Chinchilla*. Here, we test if potential gender and occupation biases manifest in unfair outcomes on coreference resolutions, using the Winogender dataset [43] in a zero-shot setting. Winogender tests whether a model can correctly determine if a pronoun refers to different occupation words. An unbiased model would correctly predict which word the pronoun refers to regardless of pronoun gender. We follow the same setup as in Rae et al. [38] (described further in Section H.3).

As shown in Table A12, *Chinchilla* correctly resolves pronouns more frequently than *Gopher* across all groups. Interestingly, the performance increase is considerably smaller for male pronouns (increase of 3.2%) than for female or neutral pronouns (increases of 8.3% and 9.2% respectively). We also consider *gotcha* examples, in which the correct pronoun resolution contradicts gender stereotypes (determined by labor statistics). Again, we see that *Chinchilla* resolves pronouns more accurately than *Gopher*. When breaking up examples by male/female gender and *gotcha/not gotcha*, the largest improvement is on female *gotcha* examples (improvement of 10%). Thus, though *Chinchilla* uniformly overcomes gender stereotypes for more coreference examples than *Gopher*, the rate of improvement is higher for some pronouns than others, suggesting that the improvements conferred by using a more compute-optimal model can be uneven.

**Sample toxicity.**   Language models are capable of generating toxic language—including insults, hate speech, profanities and threats [14, 38]. While toxicity is an umbrella term, and its evaluation in LMs comes with challenges [56, 55], automatic classifier scores can provide an indication for the levels of harmful text that a LM generates. Rae et al. [38] found that improving language modelling loss by increasing the number of model parameters has only a negligible effect on toxic text generation (unprompted); here we analyze whether the same holds true for a lower LM loss achieved via more compute-optimal training. Similar to the protocol of Rae et al. [38], we generate 25,000 unprompted samples from *Chinchilla*, and compare their *PerspectiveAPI* toxicity score distribution to that of

| Task | Chinchilla | Gopher | Task | Chinchilla | Gopher |
|---|---|---|---|---|---|
| hyperbaton | 54.2 | 51.7 | movie_dialog_same_or_diff | 54.5 | 50.7 |
| causal_judgment | 57.4 | 50.8 | winowhy | 62.5 | 56.7 |
| formal_fallacies_syllogisms_neg | 52.1 | 50.7 | movie_recommendation | 75.6 | 50.5 |
| crash_blossom | 47.6 | 63.6 | moral_permissibility | 57.3 | 55.1 |
| discourse_marker_prediction | 13.1 | 11.7 | strategyqa | 68.3 | 61.0 |
| general_knowledge_json | 94.3 | 93.9 | nonsense_words_grammar | 78.0 | 61.4 |
| sports_understanding | 71.0 | 54.9 | metaphor_boolean | 93.1 | 59.3 |
| implicit_relations | 49.4 | 36.4 | navigate | 52.6 | 51.1 |
| penguins_in_a_table | 48.7 | 40.6 | presuppositions_as_nli | 49.9 | 34.0 |
| intent_recognition | 92.8 | 88.7 | temporal_sequences | 32.0 | 19.0 |
| reasoning_about_colored_objects | 59.7 | 49.2 | question_selection | 52.6 | 41.4 |
| logic_grid_puzzle | 44.0 | 35.1 | logical_fallacy_detection | 72.1 | 58.9 |
| timedial | 68.8 | 50.9 | physical_intuition | 79.0 | 59.7 |
| epistemic_reasoning | 60.6 | 56.4 | physics_mc | 65.5 | 50.9 |
| ruin_names | 47.1 | 38.6 | identify_odd_metaphor | 68.8 | 38.6 |
| hindu_knowledge | 91.4 | 80.0 | understanding_fables | 60.3 | 39.6 |
| misconceptions | 65.3 | 61.7 | logical_sequence | 64.1 | 36.4 |
| implicatures | 75.0 | 62.0 | mathematical_induction | 47.3 | 57.6 |
| disambiguation_q | 54.7 | 45.5 | fantasy_reasoning | 69.0 | 64.1 |
| known_unknowns | 65.2 | 63.6 | SNARKS | 58.6 | 48.3 |
| dark_humor_detection | 66.2 | 83.1 | crass_ai | 75.0 | 56.8 |
| analogical_similarity | 38.1 | 17.2 | entailed_polarity | 94.0 | 89.5 |
| sentence_ambiguity | 71.7 | 69.1 | irony_identification | 73.0 | 69.7 |
| riddle_sense | 85.7 | 68.2 | evaluating_info_essentiality | 17.6 | 16.7 |
| date_understanding | 52.3 | 44.1 | phrase_relatedness | 94.0 | 81.8 |
| analytic_entailment | 67.1 | 53.0 | novel_concepts | 65.6 | 59.1 |
| odd_one_out | 70.9 | 32.5 | empirical_judgments | 67.7 | 52.5 |
| logical_args | 56.2 | 59.1 | figure_of_speech_detection | 63.3 | 52.7 |
| alignment_questionnaire | 91.3 | 79.2 | english_proverbs | 82.4 | 57.6 |
| similarities_abstraction | 87.0 | 81.8 | Human_organs_senses_mcc | 85.7 | 84.8 |
| anachronisms | 69.1 | 56.4 | gre_reading_comprehension | 53.1 | 27.3 |

Table A10: ***Chinchilla* BIG-bench results.** For each subset of BIG-bench [49], we show *Chinchilla* and *Gopher*'s accuracy.

*Gopher*-generated samples. Several summary statistics indicate an absence of major differences: the mean (median) toxicity score for *Gopher* is 0.081 (0.064), compared to 0.087 (0.066) for *Chinchilla*, and the 95$^{th}$ percentile scores are 0.230 for *Gopher*, compared to 0.238 for *Chinchilla*. That is, the large majority of generated samples are classified as non-toxic, and the difference between the models is negligible. In line with prior findings [38], this suggests that toxicity levels in unconditional text generation are largely independent of the model quality (measured in language modelling loss), i.e. that better models of the training dataset are not necessarily more toxic.

## J  Model Card

We present the *Chinchilla* model card in Table A13, following the framework presented by Mitchell et al. [35].

| Model Details | |
|---|---|
| Model Date | March 2022 |
| Model Type | Autoregressive Transformer Language Model (Section 4.1 for details) |

| Intended Uses |
|---|

| | |
|---|---|
| Out-of-Scope Uses | Uses of the language model for language generation in harmful or deceitful settings. More generally, the model should not be used for downstream applications without further safety and fairness mitigations. |

### Factors

| | |
|---|---|
| Card Prompts – Relevant Factor | Relevant factors include which language is used. Our model is trained on English data. Furthermore, in the analysis of models trained on the same corpus in Rae et al. [38], we found it has unequal performance when modelling some dialects (e.g., African American English). Our model is designed for research. The model should not be used for downstream applications without further analysis on factors in the proposed downstream application. |
| Card Prompts – Evaluation Factors | See the results in Rae et al. [38] which analyzes models trained on the same text corpus. |

### Metrics

| | |
|---|---|
| Model Performance Measures | • Perplexity and bits per byte on language modelling datasets

• Accuracy on completion tasks, reading comprehension, MMLU, BIG-bench and fact checking.

• Exact match accuracy for question answering.

• Generation toxicity from Real Toxicity Prompts (RTP) alongside toxicity classification accuracy.

• Gender and occupation bias. Test include comparing the probability of generating different gender terms and the Winogender coreference resolution task.

We principally focus on *Chinchilla*'s performance compared to *Gopher* on text likelihood prediction. |
| Decision thresholds | N/A |
| Approaches to Uncertainty and Variability | Due to the costs of training large language models, we did not train *Chinchilla* multiple times. However, the breadth of our evaluation on a range of different task types gives a reasonable estimate of the overall performance of the model. Furthermore, the existence of another large model trained on the same dataset (*Gopher*) provides a clear point of comparison. |

### Evaluation Data

| Datasets | • Language modelling on LAMBADA, Wikitext103 [34], C4 [40], PG-19 [39] and the Pile [13].

• Language understanding, real world knowledge, mathematical and logical reasoning on the Massive Multitask Language Understanding (MMLU) benchmark [16] and on the "Beyond the Imitation Game Benchmark" (BIG-bench) [49].

• Question answering (closed book) on Natural Questions [26] and TriviaQA [21].

• Reading comprehension on RACE [27]

• Common sense understanding on HellaSwag [58], PIQA [3], Winogrande [44], SIQA [45], BoolQ [10], and TruthfulQA [31]. |
|---|---|
| Motivation | We chose evaluations from Rae et al. [38] to allow us to most directly compare to *Gopher*. |
| Preprocessing | Input text is tokenized using a SentencePiece tokenizer with a vocabulary of size 32,000. Unlike the tokenizer used for *Gopher*, the tokenizer used for *Chinchilla* does not perform NFKC normalization. |

**Training Data**

The same dataset is used as in Rae et al. [38]. Differences in sampling are shown in Table A1.

**Quantitative Analyses**

| Unitary Results | Section 4.2 gives a detailed description of our analysis. Main take-aways include:

• Our model is capable of outputting toxic language as measured by the PerspectiveAPI. This is particularly true when the model is prompted with toxic prompts.

• Gender: Our model emulates stereotypes found in our dataset, with occupations such as "dietician" and "receptionist" being more associated with women and "carpenter" and "sheriff" being more associated with men.

• Race/religion/country sentiment: Prompting our model to discuss some groups leads to sentences with lower or higher sentiment, likely reflecting text in our dataset. |
|---|---|
| Intersectional Results | We did not investigate intersectional biases. |

**Ethical Considerations**

| Data | The data is the same as described in Rae et al. [38]. |
|---|---|
| Human Life | The model is not intended to inform decisions about matters central to human life or flourishing. |

| | |
|---|---|
| Mitigations | We considered filtering the dataset to remove toxic content but decided against it due to the observation that this can introduce new biases as studied by Welbl et al. [55]. More work is needed on mitigation approaches to toxic content and other types of risks associated with language models, such as those discussed in Weidinger et al. [54]. |
| Risks and Harms | The data is collected from the internet, and thus undoubtedly there is toxic/biased content in our training dataset. Furthermore, it is likely that personal information is also in the dataset that has been used to train our models. We defer to the more detailed discussion in Weidinger et al. [54]. |
| Use Cases | Especially fraught use cases include the generation of factually incorrect information with the intent of distributing it or using the model to generate racist, sexist or otherwise toxic text with harmful intent. Many more use cases that could cause harm exist. Such applications to malicious use are discussed in detail in Weidinger et al. [54]. |

Table A13: ***Chinchilla* model card.** We follow the framework presented in Mitchell et al. [35].

# K   List of trained models

In Table A14 we list the model size and configuration of all models used in this study. Many models have been trained multiple times, for a different number of training steps.

Table A11: **Closed-book question answering.** For Natural Questions [26] and TriviaQA [21], *Chinchilla* outperforms *Gopher* in all cases. On Natural Questions, *Chinchilla* outperforms GPT-3. On TriviaQA we show results on two different evaluation sets to allow for comparison to GPT-3 and to open book SOTA (FiD + Distillation [20]).

|  | Method | *Chinchilla* | *Gopher* | GPT-3 | SOTA (open book) |
|---|---|---|---|---|---|
| Natural Questions (dev) | 0-shot | 16.6% | 10.1% | 14.6% | |
|  | 5-shot | 31.5% | 24.5% | - | 54.4% |
|  | 64-shot | 35.5% | 28.2% | 29.9% | |
| TriviaQA (unfiltered, test) | 0-shot | 67.0% | 52.8% | 64.3 % | |
|  | 5-shot | 73.2% | 63.6% | - | - |
|  | 64-shot | 72.3% | 61.3% | 71.2% | |
| TriviaQA (filtered, dev) | 0-shot | 55.4% | 43.5% | - | |
|  | 5-shot | 64.1% | 57.0% | - | 72.5% |
|  | 64-shot | 64.6% | 57.2% | - | |

Table A12: **Winogender results. Left:** *Chinchilla* consistently resolves pronouns better than *Gopher*. **Right:** *Chinchilla* performs better on examples which contradict gender stereotypes (*gotcha* examples). However, difference in performance across groups suggests *Chinchilla* exhibits bias.

|  | *Chinchilla* | *Gopher* |
|---|---|---|
| All | 78.3% | 71.4% |
| Male | 71.2% | 68.0% |
| Female | 79.6% | 71.3% |
| Neutral | 84.2% | 75.0% |

|  | *Chinchilla* | *Gopher* |
|---|---|---|
| Male *gotcha* | 62.5% | 59.2% |
| Male *not gotcha* | 80.0% | 76.7% |
| Female *gotcha* | 76.7% | 66.7% |
| Female *not gotcha* | 82.5% | 75.8% |

| Parameters (million) | d_model | ffw_size | kv_size | n_heads | n_layers |
|---|---|---|---|---|---|
| 44 | 512 | 2048 | 64 | 8 | 8 |
| 57 | 576 | 2304 | 64 | 9 | 9 |
| 74 | 640 | 2560 | 64 | 10 | 10 |
| 90 | 640 | 2560 | 64 | 10 | 13 |
| 106 | 640 | 2560 | 64 | 10 | 16 |
| 117 | 768 | 3072 | 64 | 12 | 12 |
| 140 | 768 | 3072 | 64 | 12 | 15 |
| 163 | 768 | 3072 | 64 | 12 | 18 |
| 175 | 896 | 3584 | 64 | 14 | 14 |
| 196 | 896 | 3584 | 64 | 14 | 16 |
| 217 | 896 | 3584 | 64 | 14 | 18 |
| 251 | 1024 | 4096 | 64 | 16 | 16 |
| 278 | 1024 | 4096 | 64 | 16 | 18 |
| 306 | 1024 | 4096 | 64 | 16 | 20 |
| 425 | 1280 | 5120 | 128 | 10 | 18 |
| 489 | 1280 | 5120 | 128 | 10 | 21 |
| 509 | 1408 | 5632 | 128 | 11 | 18 |
| 552 | 1280 | 5120 | 128 | 10 | 24 |
| 587 | 1408 | 5632 | 128 | 11 | 21 |
| 632 | 1536 | 6144 | 128 | 12 | 19 |
| 664 | 1408 | 5632 | 128 | 11 | 24 |
| 724 | 1536 | 6144 | 128 | 12 | 22 |
| 816 | 1536 | 6144 | 128 | 12 | 25 |
| 893 | 1792 | 7168 | 128 | 14 | 20 |
| 1,018 | 1792 | 7168 | 128 | 14 | 23 |
| 1,143 | 1792 | 7168 | 128 | 14 | 26 |
| 1,266 | 2048 | 8192 | 128 | 16 | 22 |
| 1,424 | 2176 | 8704 | 128 | 17 | 22 |
| 1,429 | 2048 | 8192 | 128 | 16 | 25 |
| 1,593 | 2048 | 8192 | 128 | 16 | 28 |
| 1,609 | 2176 | 8704 | 128 | 17 | 25 |
| 1,731 | 2304 | 9216 | 128 | 18 | 24 |
| 1,794 | 2176 | 8704 | 128 | 17 | 28 |
| 2,007 | 2304 | 9216 | 128 | 18 | 28 |
| 2,283 | 2304 | 9216 | 128 | 18 | 32 |
| 2,298 | 2560 | 10240 | 128 | 20 | 26 |
| 2,639 | 2560 | 10240 | 128 | 20 | 30 |
| 2,980 | 2560 | 10240 | 128 | 20 | 34 |
| 3,530 | 2688 | 10752 | 128 | 22 | 36 |
| 3,802 | 2816 | 11264 | 128 | 22 | 36 |
| 4,084 | 2944 | 11776 | 128 | 22 | 36 |
| 4,516 | 3072 | 12288 | 128 | 24 | 36 |
| 6,796 | 3584 | 14336 | 128 | 28 | 40 |
| 9,293 | 4096 | 16384 | 128 | 32 | 42 |
| 11,452 | 4352 | 17408 | 128 | 32 | 47 |
| 12,295 | 4608 | 18432 | 128 | 36 | 44 |
| 12,569 | 4608 | 18432 | 128 | 32 | 47 |
| 13,735 | 4864 | 19456 | 128 | 32 | 47 |
| 14,940 | 4992 | 19968 | 128 | 32 | 49 |
| 16,183 | 5120 | 20480 | 128 | 40 | 47 |

Table A14: **All models.** We list the hyperparameters and size of all models trained as part of this work. Many shown models have been trained with multiple learning rate schedules/number of training tokens.