# OpenReview forum: "An empirical analysis of compute-optimal large language model training"
_NeurIPS.cc/2022/Conference — NeurIPS 2022 Accept_

### Official Review · Reviewer_1hyi · 2022-07-09

**Rating:** 8
**Confidence:** 5
**Soundness:** 4 excellent
**Presentation:** 4 excellent
**Contribution:** 4 excellent

**Summary:**

Current NLP is centered around language modeling. We've observed that their performance can significantly increase when their parameter count is increased and therefore a lot of effort has been focused on scaling that property.
The authors here note that the community hasn't really explored the importance of the amount of training data, and so that is the focus here. The authors, through many experiments, discover that when the number of parameters is doubled, so should the amount of training data.
The experiments are thorough and the marquee result is incredibly interesting and impactful- with a 70B param model they are able to outperform GPT-3 and other 170B+ parameter models on many challenging downstream tasks including MMLU and some Big Bench tasks.

**Questions:**

Are all tokens created equal? I didn't really see a discussion of this in the paper, but when you talk about doubling the training data, is any data going to be OK? Are there specific domains that are more beneficial for downstream tasks? I'm sure 20 papers could be written about this question, I don't think you need to provide all the answers, but it could be beneficial to have a brief discussion of this in the paper.



**Limitations:**

The authors adequately addressed the limitations and potential negative societal impact of their work.

**Strengths And Weaknesses:**

Strengths:
1) Very thorough experiments.
2) Good writing
3) An easy-to-understand conclusion that is very straightforward to implement.
4) The bottom line here (how much data is needed to train big models) is an incredibly important result which will be useful for years to come.
5) This paper explores a very important but very under-explored topic.

Weaknesses:
1) No code/models are released, which is really bad for reproducibility.

---

> ### Author Response · Authors · 2022-08-03
> **Thanks for the review**
>
> Thank you for your review. With respect to your question “Are all tokens created equal?” we added a brief discussion at the end of Appendix Section C. We certainly think that not all tokens are equal and a better understanding of how to assess data quality will be of utmost importance in the creation of even better language models.

---

### Official Review · Reviewer_x4Sw · 2022-07-11

**Rating:** 8
**Confidence:** 3
**Soundness:** 4 excellent
**Presentation:** 4 excellent
**Contribution:** 3 good

**Summary:**

Kaplan et al. (2020) suggested that large models should not be trained to their lowest possible loss to be compute-optimal: instead, model size should grow faster than the size of the training set, given a fixed computational budget increase.
This paper is a more thorough investigation of the question: what are the optimal model size and number of training tokens for a given training budget, tuning additional hyperparameters ignored by prior work? The authors find that model size and training budget should, in fact and in contrast to prior findings, be scaled equally -- since this isn't typically done, many state-of-the-art models are undertrained. They then use this insight to train a new model (which they call Chinchilla), which uses more data but the same compute as a state-of-the-art model and significantly outperforms the latter on multiple downstream tasks.

**Questions:**

None.

**Limitations:**

The authors do explicitly address the limitations of their work in Section 5, which is great. I don't see any direct negative societal impact of the word besides, maybe, the deployment of biased language models, but the authors mention this risk.

**Strengths And Weaknesses:**

Strengths:
- This paper provides experimental results that contradict previous findings and might be quite consequential for the future of large language models.
- The experiments are quite extensive. (Except for the fact that there are limited complete pretraining runs, but this makes sense given the computational cost.)
- The paper is clearly written.

Weaknesses:
- No obvious ones.

---

> ### Author Response · Authors · 2022-08-03
> **Thank you for the review.**
>
> Thank you for the review. Is there anything more we can do to improve the paper?

---

### Official Review · Reviewer_CUYr · 2022-07-12

**Rating:** 8
**Confidence:** 4
**Soundness:** 4 excellent
**Presentation:** 4 excellent
**Contribution:** 3 good

**Summary:**

This paper explores what is the best model size and number or tokens for training Transformer models. They trained over 400 language models with model sizes ranging from 70M to over 16B parameters on 5 to 500 billion tokens, and it suggests the model size and training data should be scaled equally. Based on this finding, they train a 70B parameters model, a predicted compute-optimal model, called Chinchilla, on more training data which outperforms the Gopher 280B model with same compute budget.

**Questions:**

I'm wondering how the data quality affects this scale-law.

**Ethics Review Area:**

["I don’t know"]

**Strengths And Weaknesses:**

Strengths:

The paper is very interesting and inspiring.

It gives a good guideline when people want to scale up their LMs.

Strong results.


Weaknesses:

It is only tested on autoregressive models. I’m wondering whether this observation is held on BERT-like models.

---

> ### Author Response · Authors · 2022-08-03
> **Thank you for the review.**
>
> Thank you for the review. Understanding the optimal scaling properties of other model types is an exciting direction of future work. With respect to the impact of data quality, in the supplement we show isoFLOP analysis on both the C4 and GitHub datasets. In both cases, we recover the same scaling exponents (~0.5) suggesting that while a model trained on higher quality data may be better, we actually expect the scaling between model size and dataset size to hold independent of data quality (though better data quality will likely lead to better model performance).

---

### Official Review · Reviewer_QHWQ · 2022-07-21

**Rating:** 7
**Confidence:** 4
**Soundness:** 3 good
**Presentation:** 4 excellent
**Contribution:** 3 good

**Summary:**

This paper presents an empirical study to explore the scaling law for training Transformer-based large language models (LLM). In this paper, the scaling law is the trade-off between model size and training tokens, and it is explored by different fitting methods. To obtain the empirical data for fitting the scaling curves, the authors train various language models by varying the model size and the training FLOP counts. The estimated power-law relationship indicates that the optimal scaling way is to increase the size of training data equally while enlarging the model size. To verify the correctness of this optimal scaling law, the authors train the *Chinchilla* model with larger training tokens but decrease the model size to 70B compared to *Gopher* (280B). According to the evaluation results shown in Appendix H, *Chinchilla* outperforms other *Gopher* on most benchmarks while sharing the same FLOPs cost.

The contributions of this paper are as follows: 1) refining the existing scaling law explored by *Gopher*; 2) revealing an optimal scaling law regarding the training tokens and model size for training LLMs.


**Questions:**

1. In Section 3.1 (Line 130), you mentioned that smoothing and interpolation methods were applied to each training loss curve. I found the details of the smoothing method in Appendix D.1. But which interpolation method did you use in Approach 1?
2. Have you examined the overlap between pre-training data and downstream fine-tuning data? (especially for the evaluation of language modeling tasks)
3. You mentioned that current LLMs are under-trained. Is there any metric to evaluate the bottleneck of model capacity instead of empirical methods?

***Update***: The authors have replied to Q1, Q2 with further clarification.

**Limitations:**

- Societal Impact: The societal impact is positive. For those researchers who work on developing large-scale language models, the authors provide a useful scaling law to save the computational budget.
- Limitation: The major limitation still lies in the research methodology, the empirical study may not always be solid as I stated in the Weaknesses part.


**Strengths And Weaknesses:**

**Strength**:
1. ***Significance***: The scaling law explored in this empirical study is useful to train the LLM. The authors also claim that the current LLMs are under-trained, the conclusion in this paper may show an interesting direction for this community to keep optimizing the LLMs: we need to pay attention to efficiently learning the data instead of enlarging the model size.
2. ***Originality***: Although this paper follows the research methodology of a previous study [1], i.e., an empirical study, this paper eventually shows a new scaling law.

**Weakness**:
1. ***Soundness***: Since this paper uses the empirical method to explore the optimal scaling law, the theoretical foundation is not very solid. How can we judge whether a model is under-trained in a more sound way? Besides, given that the three modeling approaches in this paper rely on empirical training records, one problem is the random factors of training. As the authors claimed in Checklist 3.(c), the costs of training these models are expensive, they did not perform training with different random seeds.
2. ***Clarity***: This paper is easy to follow and well-written. But there are some typos in the paper:
    - Line 1: a transformer -> a Transformer
    - Line 37: budget, Instead, -> budget, instead
    - Line 71: [24] first -> Kaplan et al. [24] first
    - Line 74: differs from [24] -> differs from Kaplan et al. [24]
    - The paper title is different from the information shown in this forum

**Reference**:

[1] Kaplan, J., McCandlish, S., Henighan, T., Brown, T. B., Chess, B., Child, R., ... & Amodei, D. (2020). Scaling laws for neural language models. arXiv preprint arXiv:2001.08361. https://arxiv.org/abs/2001.08361

***Update***: The authors addressed most of concerns mentioned in Weaknesses part.

---

> ### Author Response · Authors · 2022-08-03
> **Thank you for your review**
>
> Thank you for the thorough and careful review.  We have updated the paper to correct the typos you pointed out. With respect to the questions:
>
> * Question 1: Thank you for pointing this out. We used scipy interp1d which uses a linear interpolant between data points. We have updated the text to make this clear.
> * Question 2: For Curation Corpus, Wikitext103, and LAMBADA we did perform test set filtering. However for the Pile, MMLU, and BIG-Bench we did not perform test set filtering as they are were collected after MassiveText was collected. There is a discussion on the test set filtering in “Scaling Language Models: Methods, Analysis & Insights from Training Gopher” [2022]. In general, a better understanding of test set filtering and the impact it has in the extreme data regime is very important. However as Gopher and Chinchilla were trained on the same data we suspect the impact to be small, even though Chinchilla did see much more data. Additionally, the degree to which Chinchilla outperformed Gopher across many tasks (including BIG-Bench tasks) which are unlikely to be present in the training data suggests that the performance gains are not due to leakage.
> * Question 3 + Weakness 1: Better metrics to quantify how a model is undertrained is a very important research question, however one that we have not yet systematically approached. In the supplement, we show Figure A4 which attempts to provide some early quantification of this question. Specifically we show a model trained based on the approach from Kaplan et al compared to that which our analysis suggests. We find that a smaller model trained on more data is more performant.
> * With respect to the random seeds: for training these language models (smaller and larger), there is minimal variance between random seeds and therefore there is no benefit in running multiple seeds.

---

> > ### Comment · Reviewer_QHWQ · 2022-08-07
> > **Reply to Authors’ Response**
> >
> > Thank you for the reply. The authors’ response addressed my concern of interpolation method and data leakage.
> > Regarding the weakness part, although the authors did not study the metrics from the theoretical perspective, there are some empirical results and conclusions in this paper, which are useful for practical scenarios.
> > However, I was wondering whether if there is any research paper studying the variance between random seeds when training a LLM. It would be more convincing if the author could cite the existing findings in the Checklist part to support their claim.

---

> > > ### Author Response · Authors · 2022-08-07
> > > **Random Seeds**
> > >
> > > We have trained 5 different 1.1B models with different random data and included results in the Appendix. We have copied the text below, for ease:
> > > > We trained 5 different 1.1 billion parameter models on random subsets of the data to look at the variance in final performance.
> > > We found that the average loss achieved was 2.488 with a standard deviation amongst the 5 runs of 0.00257. Given how small the differences are, we are confident than any given run is very indicative of a model of that size.

---

> > > > ### Comment · Reviewer_QHWQ · 2022-08-08
> > > > **Reply to Authors’ Response (2)**
> > > >
> > > > Thank you for including the new results in the Appendix D.1.
> > > >
> > > > The discussion above addressed my major concerns. Thus, I am glad to increase Soundness score and overall rating.

---

### Meta-Review · Area_Chair_2wH4 · 2022-08-24

**Recommendation:** Accept
**Confidence:** Certain

**Metareview:**

Four experts reviewed this paper and they all recommended acceptance. The paper finds that current Transformer-based large language models (LLM) are significantly undertrained. This is likely to be of great interest to the AI/NLP community, as it challenges current practices and recommendations from prior work. The paper's main recommendation is that, given a increase of computation budget, model size and number of training tokens should be scaled equally. The claims of the paper are supported with an extensive amount of experimentation, including 400 language models, model sizes ranging from 70M to over 16B parameters, and amounts of data ranging from 5 to 500 billion tokens. Reviewers either listed no weaknesses or had most of their concerns addressed by the authors' responses. The main remaining limitation is that the authors couldn't release any code or data, but the work seems mostly reproducible from the paper (extensive methodological and experimental details are given in the appendix).

**Award:**

Yes

---

### Decision · Program_Chairs · 2022-09-14

Accept